# Synthesis of Needle-like CoO Nanowires Decorated with Electrospun Carbon Nanofibers for High-Performance Flexible Supercapacitors

**DOI:** 10.3390/nano14211770

**Published:** 2024-11-04

**Authors:** Xiang Zhang

**Affiliations:** Department of Energy and Power Engineering, North University of China, Taiyuan 038507, China; zhangxiang@bit.edu.cn

**Keywords:** flexible supercapacitors, carbon nanofibers, needle-like CoO

## Abstract

Needle-like CoO nanowires have been successfully synthesized by a facile hydrothermal process on an electrospun carbon nanofibers substrate. The as-prepared sample mesoporous CoO nanowires aligned vertically on the surface of carbon nanofibers and cross-linked with each other, producing loosely porous nanostructures. These hybrid composite electrodes exhibit a high specific capacitance of 1068.3 F g^−1^ at a scan rate of 5 mV s^−1^ and a good rate capability of 613.7 F g^−1^ at a scan rate of 60 mV s^−1^ in a three-electrode cell. The CoO NWs@CNF//CNT@CNF asymmetric device exhibits remarkable cycling stability and delivers a capacitance of 79.3 F/g with a capacitance retention of 92.1 % after 10,000 cycles. The asymmetric device delivers a high energy density of 37 Wh kg^−1^ with a power density of 0.8 kW kg^−1^ and a high power density of 16 kW kg^−1^ with an energy density of 23 Wh kg^−1^. This study demonstrated a promising strategy to enhance the electrochemical performance of flexible supercapacitors.

## 1. Introduction

Supercapacitors have drawn considerable research attention as they play a vital role in hybrid electric vehicles and industrial equipment [1,2,3,4,5,6]. In recent years, one of the hottest spots in this area has been to develop novel electrode materials with high energy density, high rate capability, low cost and good cycling stability. Transition metal (Ru, Fe, Co, Ni, Mn, V, etc.) oxides and hydroxides distinguish themselves among numerous materials as their variety of oxidation states facilitates charge transfer and high mass densities [7,8,9,10,11,12,13,14]. Among them, ruthenium oxide is most extensively studied in view of its ideal pseudocapacitive behavior and excellent specific capacitance (over 1300 F g^−1^) [7,8]. Nevertheless, the high cost and toxicity limit its commercialization in practical supercapacitor applications. Therefore, searching for cost-effective and nontoxic materials, such as CoO_x_ and MnO_2_ has been a focus in supercapacitors. CoO is one of the best candidates because of its high theoretical specific capacitance (∼4292 F g^−1^) [14], low toxicity and non-precious nature. Unfortunately, the electrical conductivity of CoO is poor, limiting its utilization as supercapacitor material [14,15].

An effective route to achieve high specific capacity is fabricating nanosized metal oxide materials, as nanotechnology can enhance their redox kinetics by fast ion diffusion and electron transfer. Further efforts are devoted to combining these metal oxides with conductive carbon materials [14,15,16,17,18]. One-dimensional carbon nanofibers (CNF) received extensive attention among various carbon materials, using them as chemically stable supports to enhance the electrical conductivity of metallic oxides. This heightened interest is attributed to the unique one-dimensional properties, high specific surface area and good electrical conductivity. Leveraging their outstanding mechanical properties and facile preparation, electrospun carbon nanofibers are regarded as promising conductive supports for metal oxides [19,20].

Based on the above consideration, we present a heterostructure where individual electrospun carbon nanofibers functionalized with CoO nanowires at the out surface (CoO NWs@CNF) for the first time in the usage of supercapacitors in this paper. In the designed heterostructure, not only the conductivity of CoO is improved, but also the stability was preserved during the charge–discharge cycle due to the flexibility of electrospun carbon nanofibers. Electrochemical performance indicates that the CoO NWs@CNF exhibited excellent electrochemical performance as supercapacitor materials.

## 2. Experimental

### 2.1. Materials

Polyacrylonitrile (PAN, Mw = 150,000), urea, cobalt (II) nitrate (Co(NO_3_)_2_, >99.9%), N,N-dimethylformamide (DMF, 99.8%) were purchased from Sigma-Aldrich (St. Louis, MO, USA) and used as received. All the chemicals employed were of analytical grade and were utilized directly without further purification.

### 2.2. Synthesis of Carbon Nanofibers

The carbon nanofibers were obtained by a facile electrospinning process of PAN and a subsequent annealing in Ar. Typically, 0.8 g PAN was dissolved in 10 mL DMF under stirring for 12 h. The prepared homogeneous solution was then loaded into plastic syringes (10 mL) with a needle of 22 G and subsequently placed into a commercial electrospinning setup (Electrospunra, Microtools Pvt Ltd., Singapore). A high-voltage power of 20 kV was applied to the needle tip. The flow rate of fluid was set to 1.0 mL h^−1^. The humidity level inside the electrospinning chamber was 55 ± 5%. The nanofibers were collected on aluminum foil wrapped around a flat plate placed 12 cm below the needle tip. The as-electrospun PAN nanofibers were initially stabilized at 280 °C for 3 h in air. The as-stabilized sample was carbonized in a tubular furnace at 950 °C for 1 h in Ar. The heating and cooling rate was set to be 5 °C min^−1^. After that the as-prepared CNF was as cleaned ultrasonically in 2 M HCl, washed with deionized water, and dried at 80 °C overnight.

### 2.3. Synthesis of CoO Nanowires@Electrospun Carbon Nanofibers

A hierarchical structure composed of cobalt oxide (CoO) nanowires embedded in electrospun carbon nanofibers (CoO NWs@CNF) was synthesized through a hydrothermal reaction followed by calcination. The schematic synthesis route is depicted in Figure 1. In this process, 2 mmol Co(NO_3_)_2_·6H_2_O, 10 mmol urea (CO(NH_2_)_2_) and 4 mmol NH4F were added into 70 mL deionized water and stirred for 30 min at room temperature. Subsequently, the obtained homogeneous solution was transferred into a Teflon-lined stainless steel autoclave with a piece of CNF (20 mm × 30 mm), immersed into the reaction solution for hydrothermal process. The upper side of CNF was protected by a polytetrafluoroethylene tape. The Teflon-lined stainless steel autoclave was then placed in an oven at 120 °C for 8 h. After naturally cooling down to room temperature, the sample was collected via centrifugation, washed with deionized water and alcohol to remove impurities. Finally, the sample was annealed in Ar at 350 °C for 2 h to prepare the CoO NWs@CNF. 

### 2.4. Characterization

X-ray diffraction measurement was conducted using a Bruker AXS D8 Advance X-ray diffractometer (Billerica, MA, USA) equipped with Cu Kα radiation within the range of 20° to 80°. Raman spectra were recorded using a Dilor model OMARS 89-Z24 microprobe spectrometer, with excitation provided by an Ar+ ion laser of 514.5 nm. The morphological features and chemical composition were examined with a field emission scanning electron microscope (FE-SEM, JEOL-6701F, Tokyo, Japan) using a JEOL-6701F instrument. 

### 2.5. Electrochemical Measurements

Electrochemical measurements were conducted using a CHI 760D Electrochemical Workstation (Bee Cave, TX, USA) with 1 M KOH as the electrolyte. The electrospun carbon nanofibers-supported hybrid nanostructure (approximately 6 cm^2^ area; CoO mass: approximately 3.0 mg cm^−2^) directly served as the working electrode. A Pt plate and Ag/AgCl were employed as the counter electrode and the reference electrode respectively, with all potentials referenced to the Ag/AgCl electrode. The cyclic voltammograms were recorded from −0.1 to 0.45 V at different scan rates ranged from 5 to 60 mV s^−1^. The galvanostatic charge-discharge tests were performed at different current densities from −0.1 to 0.45 V. The cyclic stability was evaluated by the current density of 2 A g^−1^ for over 5000 cycles, which were acquired using an Arbin testing system (MSTAT, Arbin Instruments, College Station, TX, USA).

## 3. Results and Discussion

The schematic representation of the procedure employed to fabricate needle-like CoO nanowires on carbon nanofibers is illustrated in Figure 1. The carbon nanofibers were used as current collectors to grow the CoO nanowires array. Firstly, urea underwent hydrolysis in the reactor and produced a large amount of CO_3_^2−^ and OH^−^ gradually. When the concentration of CO_3_^2−^ and OH^−^ anions increases to a certain extent, further reactions lead to the formation of cobalt carbonate nuclei on the surface of carbon nanofibers. As the reaction progressed, the crystal growth of nuclei began to impinge on other adjacent crystals and preferentially assembled along a specific direction. Consequently, large-scale uniform cobalt hydroxide carbonate nanowires were formed on the carbon nanofibers directly. Secondly, the cobalt hydroxide carbonate nanowires array was completely decomposed and converted to a CoO nanowires array under the annealing process. The overall chemical reaction can be described by the following two equations:Co^2+^ + 0.5CO_3_^2−^ + OH^−^ + nH_2_O →Co(OH)(CO_3_)_0.5_ · nH_2_O↓(1)
Co(OH) (CO_3_)_0.5_ · nH_2_O → CoO + 0.5CO_2_↑ + (n + 0.5)H_2_O(2)

To confirm the crystallographic structure of the as-synthesized needle-like CoO NWs@CNF, the X-ray diffraction pattern was conducted and shown in Figure 2a. The diffraction peaks at 36.6°, 42.6°, 61.6°, 73.9° and 77.6° were well-indexed to the (111), (200), (220), (311) and (222) planes of face-centered cubic CoO (JCPDS card no. 48-1719). Those sharp peaks verified that cobalt precursors were transformed into highly crystalline CoO successfully through an annealing process. The XRD pattern of carbon nanofibers was depicted in Appendix A. Two broad peaks correspond to the (002) and (100) planes of graphite. The TGA data suggest that the mass loading of CoO nanowires’ active materials is ∼68.2 wt% (Appendix A).

The Raman spectra of CoO NWs@CNF between 500 and 3500 cm^−1^ are shown in Figure 2b. The broadened characteristic frequencies at 1357.6 and 1579.7 cm^−1^ correspond to the D band and G band of the carbon matrix, respectively [21,22,23]. The D-band (disorder-induced phonon mode) can be attributed to defects and disordered portions of carbon (sp^3^-coordinated), whereas the G-band (graphite band) is indicative of ordered graphitic crystallites of carbon [23,24,25,26]. The intensity ratio of D and G bands (I_D_/I_G_) provides useful information about the number of carbon defects: a higher intensity ratio means a higher concentration of defects in the graphite comprising sp^2^ hybridization [21,25,27,28]. The intensity ratio I_D_/I_G_ of our CoO NWs@CNF was found to be 1.08, indicating a relatively high amount of disordered sections and defects.

In order to measure the porosity of CoO NWs@CNF and carbon nanofibers, the specific surface area and the pore size distribution were tested by N_2_ adsorption–desorption isotherms. Type IV isotherms with H4 hysteresis loops indicate the mesoporous structure of CoO NWs@CNF in Figure 2c,d. Appendix A show the N_2_ adsorption–desorption isothermal and BJH pore size distribution curve of carbon nanofiber. The BET-specific surface area, total pore volume and average pore size of CoO NWs@CNF were calculated to be 75.6 m^2^ g^−1^, 0.121 cm^3^ g^−1^ and 15.5 nm, respectively. As for pristine carbon nanofibers, a typical IV isotherm was clearly exhibited in Figure 2c, indicating the existence of mesopores in this material. The BET-specific surface area and total pore volume were calculated to be 95.2 m^2^ g^−1^ and 0.0155 cm^3^ g^−1^, respectively. The hierarchical porous structure of CoO NWs@CNF may be beneficial to rapid mass transfer, accommodation of volume variation, the infiltration of the electrolyte, the reduction of the lithium diffusion distance and the improvement of the electrochemical properties.

The scanning electron microscopy (SEM) of as-prepared carbon nanofibers is shown in Appendix A. The CNF derived from polyacrylonitrile (PAN) nanofibers exhibited a continuous and randomly oriented structure, and the surface of the CNF was smooth. The interconnected carbon nanofibers not only formed an intrinsic network for electron transportation but also contributed to favorable mechanical strength, allowing for direct use as a flexible substrate [29,30,31,32]. Additionally, the length of carbon nanofibers was approximately 100 µm. The morphological features of the as-synthesized needle-like CoO NWs@CNF were heterostructure (Figure 3a,b). Each carbon nanofiber was uniformly covered with numerous highly porous CoO nanowires. More magnified SEM images are displayed in Figure 3c, confirming the tidy growth of abundant CoO nanowires on the surface of the carbon nanofibers, showcasing a regular needle-like shape. These needle-like CoO nanowires have an average bottom diameter of about 50 nm and a tip diameter of approximately 20 nm, with lengths extending up to 5 µm. It clearly exhibited that the CoO nanowires stand hierarchically with many spaces in between, which would increase the ion diffusion and electron transfer during charge–discharge process [33,34,35]. Energy-dispersive X-ray spectroscopy (EDS) was performed to investigate the elements’ distribution throughout the nanostructure (Figure 3d–g). It revealed that the molar ratio of Co and O is about 1:1 for the CoO NWs@CNF, which is in good agreement with the stoichiometric ratio of CoO. The corresponding EDS mapping images for the elements of C, O and Co clearly illustrate a homogeneous distribution of CoO nanowires throughout the carbon nanofibers.

XPS spectra of CoO NWs@CNF were shown in Figure 3h,i. The Co 2p spectrum was composed of two well-defined single peaks at 776.4 eV and 794.5 eV and two shakeup satellite peaks (782.8 and 802.5 eV), while the weak peaks corresponded to Co^2+^ [36,37,38]. The O1s spectra were compromised of one sharp peak and a shoulder. The sharp peak around 529.6 eV ascribed to the presence of a Co-O bond, while the shoulder at 531.2 eV corresponded to oxygen defect sites [39]. The XPS results consist of an XRD pattern, which indicates the CoO nanowires were grown on the carbon nanofibers substrate.

The electrochemical performance of the CoO NWs@CNF is evaluated in a three-electrode system. Typical cyclic voltammetry (CV) curves of the CoO NWs@CNF in the potential range of −0.10 V to 0.45V vs. Ag/AgCl with various sweep rates ranging from 5 to 60 mV s^−1^ in the electrolyte of 1M KOH were presented in Figure 4a. A pair of well-defined redox peaks within the potential range for all sweep rates, which is mainly associated with the Faradaic redox reactions related to Co-O/Co-O-OH and Co-O-OH/CoO_2_, clearly indicated the pseudocapacitive characteristics of the CoO NWs@CNF electrode. Similar shapes were observed for all the CV curves while the current density went up with increasing sweep rate. This revealed the high-power characteristics of the electrode. The pair of redox peaks shifted from 0.16 and 0.28 V to 0.12 to 0.35 V while the sweep rate increased from 5 mV s^−1^ to 60 mV s^−1^. There is only 0.04 and 0.07 V change, even with a 12-fold increase in the sweep rate. It suggested the low resistance of the carbon nanofibers electrode due to the good contact between the CoO nanowires and the carbon nanofibers. The peak current and the square root of the scan rate were calculated and plotted in Appendix A. The peak current is proportional to the square root of the scan rate, which real that the CoO NWs@CNF has very good electrochemical properties. The faradaic redox reactions of CoO nanowires in the alkaline electrolyte can be described as follows:CoO + OH^−^ ↔ CoOOH + e^−^(3)
CoOOH + OH^−^ ↔ CoO_2_ + H_2_O + e^−^(4)

The specific capacitance can be calculated from CV by the following equation:(5)$$C_{\mathrm{sp}}=\frac{1}{mv(V_{b}-V_{a})}\int_{V_{a}}^{V_{b}}I(V)dV$$
where C*_sp_* is the specific capacitance, *m* is the mass of the active materials, *v* refers to the scan rate, *V_b_* and *V_a_* refer to the upper and lower potentials of the potential window in CV and *I* (*V*) is the instant current on CV curves. The C*_sp_* is calculated to be 1068.3, 931.9, 797.5, 678.7 and 613.7 F g^−1^ at the scan rate ranging from 5 to 60 mV s^−1^ in Figure 4b. The capacitance retention is calculated to be 57.4%, with a 12 times increase in the scan rate.

The charge–discharge performances between −0.1 V and 0.45 V vs. Ag/AgCl at various current densities ranging from 1 to 20 mA g^−2^ were carried out and are shown in Figure 4c. Nearly symmetric shapes of charging and discharging curves were displayed for all current densities, indicating the high charge–discharge columbic efficiency of the CoO NWs@CNF. The discharge current exhibits an inverse relationship with the discharge time. The corresponding specific capacitance is calculated based on the data in Figure 4d by the formula C*_sp_* = IΔt/mΔV, where I is the discharge current, Δt is the discharge time, m is the mass of active materials and ΔV is the voltage range. The specific capacitance is 905.8, 748.3, 635.1, 518.2, 479.6, 421.3, 369.7 F g^−1^ at the discharge current densities of 1, 2, 4, 8, 10, 15 and 20 A g^−1^, respectively. This value (635.1 F g^−1^ at a current density of 4 A g^−1^) is far superior to that of 307 F g^−1^ at a current density of 4.55 A g^−1^ for pure CoO nanoporous walls [40] and that of 191 F g^−1^ even at a ten times lower current density (0.1 A g^−1^) for CoO-doped porous graphitic carbon microspheres [17]. This CoO NWs@CNF electrode is apparently superior to many other CoO nanostructured electrodes, as can be seen from Appendix A [41,42,43,44,45]. It strongly supported that needle-like CoO nanowires grown on carbon nanofibers’ core-shell structure play a positive role in the overall supercapacitor performance. Co(NO_3_)_2_·6H_2_O precursor with difference concentration of 0.5 mmol, 1 mmol, 2 mmol, 3 mmol and 4 mmol in hydrothermal process was studied for comparison. The specific capacitance of the four samples at a current density of 1 A g^−1^ was illustrated in Appendix A. The sample of Co(NO_3_)_2_·6H_2_O precursor with a concentration of 2 mmol delivered the best specific capacitance of 905.8 F g^−1^. Appendix A displays the galvanostatic charge and discharge performance of CoO NWs@CNF, CoO nanowires and carbon nanofibers at the current density of 1 A g^−1^. The specific capacitance of CoO NWs@CNF was better than that of CoO nanowires and carbon nanofibers at the discharge current densities of 1, 2, 4, 8, 10, 15, 20 A g^−1^, respectively (Appendix A), because the carbon nanofibers matrix enhances the intrinsic electrical conductivity and facilitates the electron transportation between the composite materials.

Appendix A shows Nyquist plots of CoO NWs and CoO NWs@CNF electrodes for comparison. The internal resistance of CoO NWs and CoO NWs@CNF electrodes are 1.12 Ω and 0.73 Ω. The reduced resistance can be attributed to the carbon nanofibers matrix, which provides a fast pathway for electron transport. Appendix A shows the SEM image of CoO NWs@CNF after 10,000 cycles. Since the carbon nanofibers matrix has good mechanical properties and the CoO NWs@CNF hierarchical structure is unique, it still can maintain the structure integrity.

The CV curves of CoO NWs@CNF and CNT@CNF were exhibited with the potential range of −0.2 V to 0.45 V and −1.2 V to 0 V at a scan rate of 5 mV s^−1^ in Appendix A, respectively. Both CV curves showed complementary potential windows, indicating the excellent potential of both materials for asymmetric supercapacitor devices with a voltage window of 0–1.6 V in KOH. A series of CV curves of the asymmetric supercapacitor device at a scan rate from 5 to 60 mV s^−1^ are shown in Appendix A. All the curves exhibited a quasi-rectangular geometry even at a high scan rate of 60 mV s^−1^ and the redox peak currents increased with the increasing scan rate. It indicated the good, fast charge–discharge properties of the device. Moreover, the galvanostatic charge–discharge performance was measured at various current densities from 1 to 20 A g^−1^ in Figure 4c. The discharge curves were nearly linear, while the whole galvanostatic charge–discharge curves were triangular in shape, suggesting a rapid I–V response and good electrochemical reversibility of the asymmetric device. The rate capability of the asymmetric device is illustrated in Appendix A. The specific capacitances were 102.6, 96.9, 86.1, 75.3, 70.1 and 68.5 F g^−1^ at 1, 2, 5, 10, 15 and 20 A g^−1^, respectively. A total of 67% of the initial specific capacitance was maintained even with a 20 times increase in current density.

The cycling stability of the asymmetric device was conducted at a current density of 5 A/g for 10,000 cycles. The CoO NWs@CNF//CNT@CNF asymmetric device exhibits remarkable cycling stability and delivers a capacitance of 79.3 F/g with a capacitance retention of 92.1% after 10,000 cycles in Figure 5a. And only a slight decrease of charge–discharge time was observed in the 1st and 10,000th cycle (inset in Figure 5a). Cycling stability is highly related to the structural stability of the CoO NWs@CNF electrodes.

The energy and power densities (E and P) were calculated from the equations $E=\int_{0}^{∆t}IV\left(t\right)dt$ and P=E/∆t, respectively, with I being the discharging current, V being the discharging voltage, dt being the time differential and Δt being the discharge time. Regone plots of the CoO NWs@CNF//CNT@CNF asymmetric device and other reported materials were shown in Figure 5b to further evaluate the performance. The CoO NWs@CNF//CNT@CNF asymmetric device delivered a high energy density of 37 Wh kg^−1^ with a power density of 0.8 kW kg^−1^ and high power density of 16 kW kg^−1^ with an energy density of 23 Wh kg^−1^, which were higher than those of most nickel and cobalt-based asymmetric supercapacitors including NiCo_2_S_4_//AC [46], CoO nanocubes//AC [47], Co_3_O_4_ nanowires//N-rGO [48], Co_3_O_4_/N-CNO//AC [49], Co_3_O_4_-C//AC [50] asymmetric supercapacitors and lead acid batteries [51]. These results indicated that CoO NWs@CNF//CNT@CNF is a promising candidate as an asymmetric supercapacitor device.

## 4. Conclusions

In summary, CoO NWs@CNF was successfully fabricated and valued in the application of pseudocapacitive performance. A high specific capacitance of 1068.3 F g^−1^ at a scan rate of 5 mV s^−1^, good rate capability and superior cycling stability were observed due to the synergetic effect of the CoO nanowires and conductive electrospun carbon nanofibers core-shell structure. Firstly, the interconnected carbon nanofibers matrix enhances the intrinsic electrical conductivity and facilitates the electron transportation between carbon nanofibers and CoO nanowires. Secondly, the needle-like CoO nanowires directly and uniformly grown on carbon nanofibers with robust adhesion offer rich active sites and speed up the Faradaic redox reaction. Thirdly, the ultrathin porous CoO nanowires array offers large open space between each nanowire, facilitates the penetration of electrolyte into the 3D electrode and provides hierarchical pathways for the ion transport. Our work presents a cost-effective hybrid material with promising pseudocapacitive performance as a potential energy storage electrode material in practical application.

## Figures and Tables

**Figure 1 nanomaterials-14-01770-f001:**
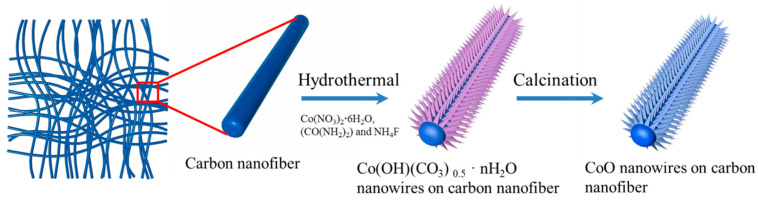
Schematic representation of the procedure used to fabricate needle-like CoO NWs@CNF.

**Figure 2 nanomaterials-14-01770-f002:**
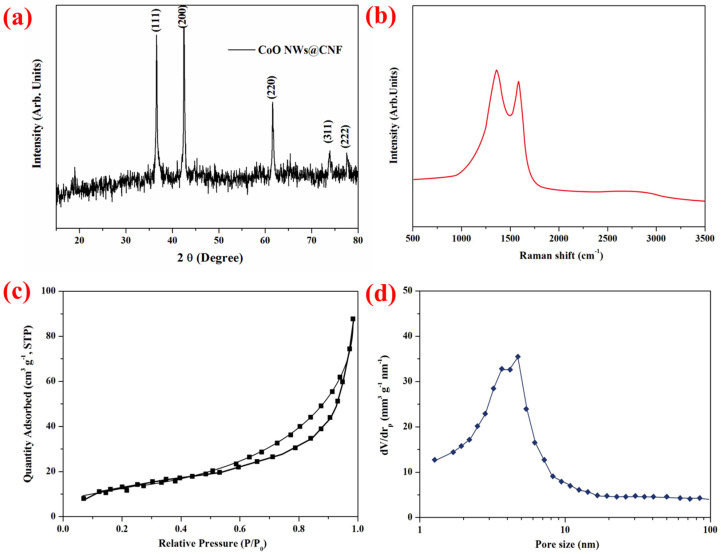
(**a**) X-ray diffraction pattern of CoO NWs@CNF. (**b**) Raman spectra of CoO NWs@CNF. (**c**) N_2_ adsorption and desorption isotherm of CoO NWs@CNF. (**d**) Pore size distribution of CoO NWs@CNF.

**Figure 3 nanomaterials-14-01770-f003:**
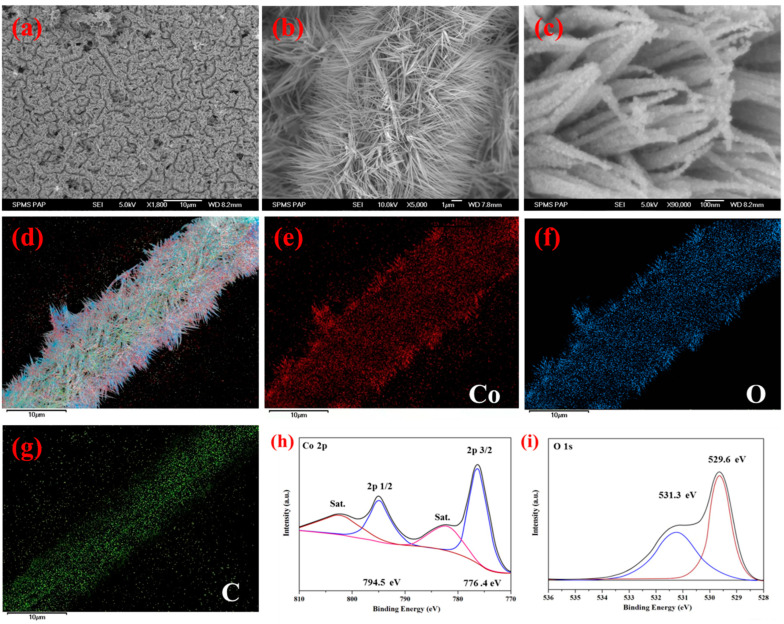
(**a**) SEM image of CoO nanowires on carbon nanofibers membrane. (**b**) SEM image of CoO Nanowires on one single carbon nanofiber. (**c**) High magnification SEM images of CoO nanowires. (**d**) SEM image of CoO NWs@CNF with elements mapping; (**e**–**g**) element mappings of cobalt (red), oxygen (blue) and carbon (green), respectively. (**h**,**i**) XPS of CoO NWs@CNF.

**Figure 4 nanomaterials-14-01770-f004:**
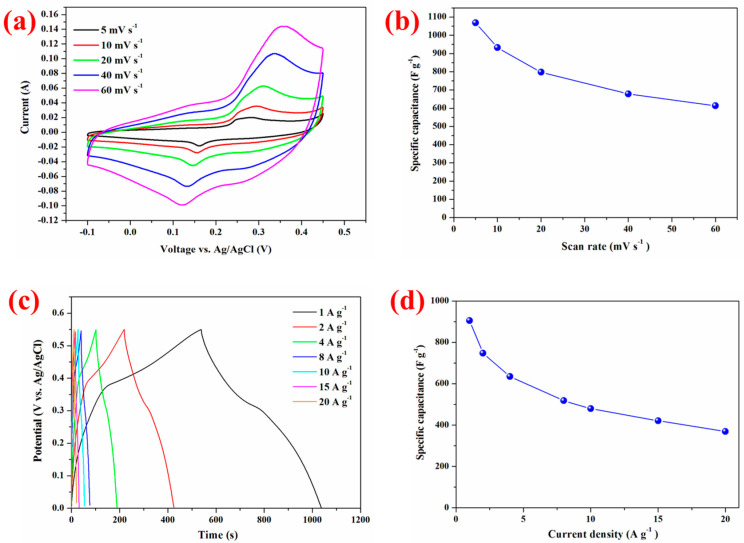
(**a**) Cyclic voltammetry measurement of CoO NWs@CNF at various scan rates ranging from 5 to 60 mV s^−1^. (**b**) Specific capacitance derived from the CV measurements. (**c**) The galvanostatic charge and discharge voltage profiles of CoO NWs@CNF at various current densities ranging from 1 to 20 A g^−1^ and (**d**) the corresponding specific capacitance as a function of current density.

**Figure 5 nanomaterials-14-01770-f005:**
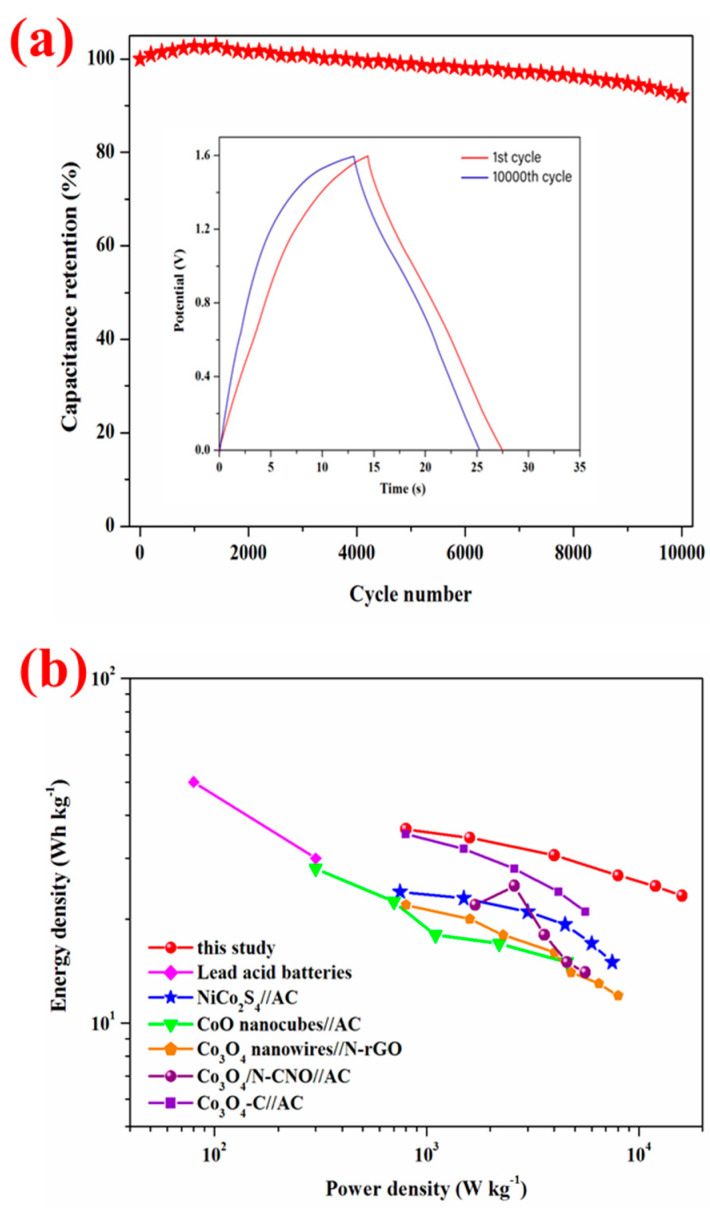
(**a**) Cycling performance for 10,000 cycles. The inset is galvanostatic charge–discharge curves at 1st and 10,000th cycles. (**b**). The Ragone plot of CoO NWs@CNF//CNT@CNF asymmetric supercapacitors and other asymmetric supercapacitors reported in the literature.

## Data Availability

Data are contained within the article.

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
