# Peer review of "Synthesis of Needle-like CoO Nanowires Decorated with Electrospun Carbon Nanofibers for High-Performance Flexible Supercapacitors"

_nanomaterials, 2024, doi:10.3390/nano14211770_

Round 1

Reviewer 1 Report

Comments and Suggestions for Authors

The article, “Synthesis of Needle-like CoO Nanowires Decorated with Electrospun Carbon Nanofibers for High Performance Flexible Supercapacitors” presents the synthesis of CoO electrode material and its characterization and application for energy storage application. The manuscript lacks many details and clarifications. Therefore, I recommend the major revision of this manuscript before publication to this journal.

1.     The authors should highlight the novelty of the work.

2.     The authors should provide more characterizations data like XPS, BET and TGA to support their results.

3.     The energy and power density should be calculated based on the two-electrode setup. These electrochemical data are missing. The authors should provide details about the device fabrications and its electrochemical performance. 

Comments on the Quality of English Language

Minor editing of English language required

Reviewer 2 Report

Comments and Suggestions for Authors

In this work, authors reported the Needle-like CoO nanowires have been successfully synthesized by a facile hydrothermal  process on electrospun carbon nanofibers substrate. The hybrid composite electrodes exhibit a high specific capacitance and electrochemical performance. However, this manuscript cannot be published in Nanomaterials before following revisions.

1.       There are several work based on metal oxide nanostructure grown on the surface of electrospun carbon fiber. How is it differ from them. Deep literature review is necessary with reference of following articles:

doi.org/10.1016/j.cej.2021.132345XRD patterns of pure CNF and CoO also should be presented and corresponding JCPDS should be uploaded.

2.       The FESEM images presented in Figure 3 and B, C as well as d do not coincide why?

3.       Detail physicochemical characterization including HR-TEM and XPS is strongly recommended.

4.       Electrochemical properties of pure CNF and CoO also should be presented. To understand the practical application of the electrode materials, real time supercapacitors should be tested and uploaded in the revised manuscript. The energy density and power density should be calculated and compared with previously reported articles.

Comments on the Quality of English Language

Extensive editing of English language required

Reviewer 3 Report

Comments and Suggestions for Authors

This short manuscript shows the preparation of CoO nanowires fabricated on CNFs and evaluation of the capacitances. Preparation of noble nanomaterials and seeking high performances are pretty important in nanosciences and I thus believe that this work is worth to be published, but I request the authors to consider the following points to improve and expand the work.

1.       The authors proposed the core-shell structures of CoO and CNTs, but there is not a normal core-shell type material. Some explanations are probably needed.

2.       There is probably no doubt to form needle-like CoO nanowires on CNFs, but elemental analysis is needed to propose it, for example using EDS mapping.

3.       Evaluate the surface areas of the composite and CNFs, and the capacitance for CNF itself to compare with the composite to insist on the higher performance.

4.       Discuss the reason why the higher performance was obtained from analyses of structures and/or chemical compositions.

5.       Here is only one sample prepared, but amount/size dependences of nanowires are needed, because many works on related composites have been done already.

Reviewer 4 Report

Comments and Suggestions for Authors

The manuscript "Synthesis of Needle-like CoO Nanowires Decorated with Electrospun Carbon Nanofibers for High Performance Flexible Supercapacitors" reports the synthesis and preliminary studies on CoO nanowires@carbon nanofibers. I suggest the authors revise the following minor and major points:

1.     In the introduction, the authors wrote: “The as-prepared sample mesoporous NiCo2O4 nanosheets”. This compound is neither presented nor discussed in the current manuscript.

2.     On page 4, Eqs. (1) and (2) need the physical states written when applicable. Please make sure to number all equations. On Page 2, two other equations are not numbered and need the physical states written.

3.     Page 3: “The interconnected carbon nanofibers not only formed an intrinsic network for electron transportation but also contributed to favorable mechanical strength, allowing for direct use as a flexible substrate.”. Please add some references to support the suggestion that the mechanical strength will be improved.

4.     Page 3: “It clearly exhibited that the CoO nanowires stand hierarchically with many spaces in between, which would increase the ion diffusion and electron transfer during charge-discharge process.”. Please add some references to support these benefits.

5.     Page 4: “A pair of well-defined redox peaks within the potential range for all sweep rates, which is mainly associated with the Faradaic redox reactions related to Co-O/Co-O-OH and Co-O-OH/CoO2, clearly indicated the pseudocapacitive characteristics of the CoO NWs@CNF electrode.”. Can the authors perform an XPS analysis to discuss the different oxidation states of Co?

6.     Electrochemical impedance spectroscopy (EIS) could have been performed to discuss the contribution of charge-transfer processes to the overall performance.

7.     Is it possible to add SEM images after the cycling tests to discuss the evolution of morphology during experiments?

8.     There is a lack of a control sample of CoO produced by conventional routes, i.e., with a granular morphology, to understand the benefits of using nanowires made of CoO, rather than just a comparison with the literature.

After answering these points, the manuscript can be published.

Comments on the Quality of English Language

The manuscript needs a revision of the English language. Please check the entire manuscript for missing grammatical and syntax errors.

Round 2

Reviewer 1 Report

Comments and Suggestions for Authors

The manuscript can be published at the current stage.

Reviewer 2 Report

Comments and Suggestions for Authors

Accept.

Reviewer 3 Report

Comments and Suggestions for Authors

The authors answered all my queries and I now consider that this manuscript can be published as it is.

Reviewer 4 Report

Comments and Suggestions for Authors

The manuscript has been properly revised. It can be accepted.